# Forensic age estimation with Bayesian convolutional neural networks based on panoramic dental X-ray imaging

**Walter de Back**[1]                    WALTER.DEBACK@TU-DRESDEN.DE
**Sebastian Seurig**[1]                SEBASTIAN.SEURIG@TU-DRESDEN.DE
**Sebastian Wagner**[1]            SEBASTIAN.WAGNER3@TU-DRESDEN.DE
**Birgit Marré**[3]              BIRGIT.MARRE@UNIKLINIKUM-DRESDEN.DE
**Ingo Roeder**[1]                        INGO.ROEDER@TU-DRESDEN.DE
**Nico Scherf**[1,2]                      NICO.SCHERF@TU-DRESDEN.DE

[1] *Institute for Medical Informatics and Biometry (IMB), Carl Gustav Carus Faculty of Medicine, TU Dresden, Dresden, Germany.*

[2] *Max Planck Institute for Human Cognitive and Brain Sciences (MPI-CBS), Leipzig, Germany.*

[3] *Department of Prosthetic Dentistry, University Hospital Carl Gustav Carus, TU Dresden, Dresden, Germany.*

**Editors:** Under Review for MIDL 2019

## Abstract

Forensic age estimation is the medical assessment of age in individuals for legal purposes. In current practice, medical experts use scoring-based methods to assess the most probable age and provide a minimum age estimate, which requires expensive training and is known to have poor inter-rater reliability. As an initial step towards a more objective, quantitative age estimation system, we use Bayesian convolutional neural networks to perform age and uncertainty estimation using a large data set of $12,000$ panoramic radiographs of the upper and lower jaws, orthopantomograms (OPTs), one of the most accurate indicators of age in subadults. Our system achieves a concordance correlation coefficient $ccc = 0.91$ on the validation set. Importantly, our method provides quantitative estimation of prediction uncertainty, which is imperative within a legal context.

## 1. Introduction

Medical assessment of age can be requested by courts and governmental agencies in legal procedures when an individual's age is unknown, especially in cases where there is doubt about legal majority (Schmeling et al., 2016). The age is evaluated by a panel of experts based on a medical examination, typically involving radiographic imaging of the hand and wrist bones and panoramic X-rays of the jaws, orthopantomograms (OPTs). OPTs are used to assess the dental mineralization status which is widely considered the most accurate indicator for age in sub-adults. Evaluation of OPTs are based on written or pictorial scoring systems such as Demirjian's method (Demirjian et al., 1973) that are time-consuming, suffer from subjectivity and lack quantitative measures of uncertainty. In an effort to automate, standardize and objectify forensic age estimation, we developed a deep learning based system for dental age estimation based on Bayesian convolutional neural networks (CNN). While previous studies have used deep learning for pediatric age estimation based on

hand and wrist bone radiographs (Spampinato et al., 2017; Halabi et al., 2018), predicting age from dental X-ray images has not received much attention. In addition, since we focus on application in a legal rather than pediatric context, here, we concentrate on the quantification of predictive uncertainty using Bayesian CNNs.

## 2. Methods

**Image data** We collected $> 12,000$ anonymized orthopantomograms (OPTs) of patients in the age 5-25 years old, imaged at the dental medicine center (UZM) at the University Hospital Dresden (UKD) over the past 15 years. Images were annotated with the patient's chronological age in month at the time of acquisition. All images were scaled down 4-fold to a size of $(h, w) = (320, 610)$ and the intensities normalized to $[0, 1]$. We used 25% of the data for validation.

**Bayesian CNN** We formulated the age estimation problem as a regression task and designed a CNN in which we used the InceptionV3 architecture (Szegedy et al., 2016) as a feature extractor, followed by two fully connected layers (with 1024 and 512 neurons with ReLU activation, resp.) after the global average pooling layer. Following (Kendall and Gal, 2017), we used two output heads to predict the most probable age and its associated aleatoric uncertainty (using resp. linear and softplus activation) for which we assumed a Gaussian likelihood with the loss function $\mathcal{L}(\theta) = \sum_i \frac{1}{2} e^{-s} \|y_i - \hat{y}_i\|^2 + \frac{1}{2} s_i$ where $\theta$ are the model parameters, $\hat{y}_i$ and $s_i = \log \hat{\sigma}^2$ are the predicted age and its log variance for sample $i$ predicted by the network. To estimate epistemic uncertainty, we use inference-time dropout as a practical tools for approximate inference using $T = 20$ stochastic forward passes to sample from the posterior distribution.

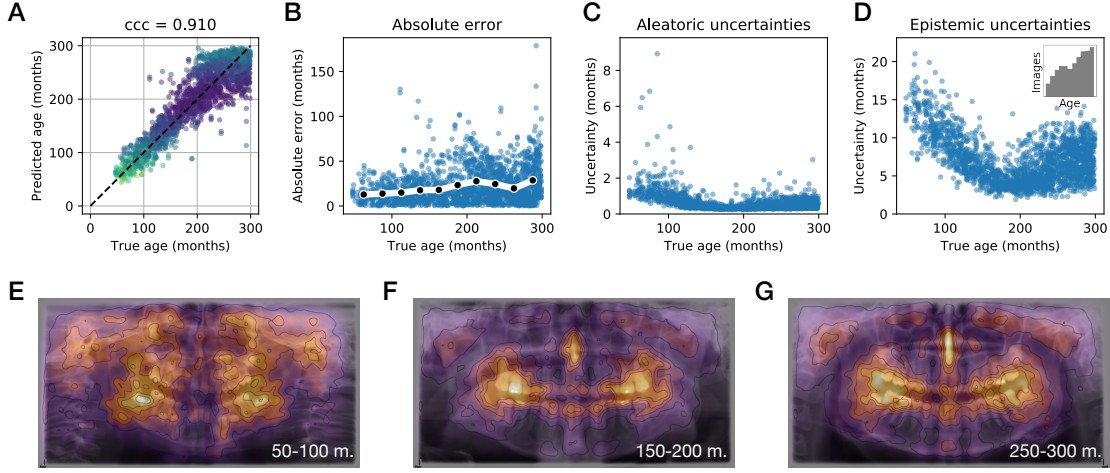

Figure 1: (A) Correlation between true and predicted age. (B) Absolute error, black points indicate MAE in various age groups. (C,D) Aleatoric and epistemic uncertanties. (E-G) Saliency maps for three different age groups.

**Training** The model was trained over 80 epochs using stochastic gradient descent with a batch size of 16 images under the adaptive moments estimator (Adam) optimizer, with horizontal mirroring as the only data augmentation strategy.

## 3. Results

After training, we obtained a concordance correlation coefficient $ccc = 0.910$ on the validation set consisting of $2,400$ images covering the whole age range (Figure 1A), representing a considerable agreement between predicted and true chronological age. However, the mean absolute error over the whole validation set was $MAE = 21.0$ months, almost two years, which is an unacceptable range for legal purposes. The $MAE$ differed significantly between age groups, with lowest errors for the youngest group $MAE_{50-75} = 12.8$ and largest errors for the young adults $MAE_{275-300} = 28.6$ (Fig. 1B).

Our Bayesian CNN allows us to quantify the predictive uncertainty in terms of aleatoric and epistemic uncertainties. Both types of uncertainty showed a non-monotonic behavior and are minimized around 180 months (15 years old) (Fig. 1C and D). The fact that uncertainty increases for older subjects may be related to the human lifespan of human dentition: less markers are available to indicate dental development near adulthood. The increased epistemic uncertainty observed in the low age range can be related to relatively low amount of training images in this range, as shown in the inset of Fig. 1D. The outliers in aleatoric uncertainty in the lower age range are due to imaging artifacts, as indicated by visual inspection of the respective images.

To explore what regions of the input image are important for the age regression, we generated saliency maps for different age ranges. Specifically, we used DeepLIFT (Shrikumar et al., 2017; Ancona et al., 2017) to construct maps for 50 randomly chosen images within different age groups and visualized the mean of these saliency maps. As shown in Fig. 1E-F, the most informative regions for the model prediction are localized around the molars, in line with our expectations. However, in the youngest age range (Fig. 1E), we see the model takes, apart from the teeth, also e.g. the maxillary sinus into account. Also unexpectedly, we observe that the nasal septum seems to be used as a marker for the oldest age range (Fig. 1G).

## 4. Conclusion

We provide a proof-of-concept for the use of Bayesian CNNs as an automated age estimation system for panoramic dental radiographs. Our system not only provides age estimates but also quantitative measures of uncertainty as well as explanatory saliency maps. Initial results are encouraging although the accuracy is not yet at the level that warrants routine application. To improve accuracy and reduce epistemic uncertainty, we next intend to use multi-task learning or curriculum learning, combining classification and regression, and employ ideas from active learning such as uncertainty-based acquisition functions. The intriguing patterns in the saliency maps are worth exploring further and could point to new potential markers for age estimation from OPTs.

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
