# OpenReview forum: "Forensic age estimation with Bayesian convolutional neural networks based on panoramic dental X-ray imaging"
_MIDL.io/2019/Conference/Abstract — MIDL Abstract 2019_

### Official Review · AnonReviewer2 · 2019-04-30
**Application of existing method to novel problem, with solid evaluation.**

**Rating:** 4
**Confidence:** 3

**Review:**

This abstract proposes to apply Bayesian CNNs [Kendall et al.] to automated forensic age estimation based on an interesting dataset of dental X-ray images. The authors offer a thorough evaluation of prediction accuracy, and the different kinds of uncertainties estimated by this methods and also interpret the predictions with saliency maps. The authors honestly discuss their results, concluding that the mean absolute error of 21 months is not sufficient for legal purposes, but offering ways forward to improve the prediction.

What do the authors make of the fact that absolute errors per age and the uncertainties per age (Fig. 1 B, C, D) are somewhat negatively correlated? Does this limit the usefulness of those uncertainty measures in practice?

---

### Official Review · AnonReviewer1 · 2019-05-02

**Rating:** 2
**Confidence:** 2

**Review:**

The paper explains a methodology to forecast the age from dental x-ray.  But I don’t see much innovation in the methodology and the results is not ideal.
I would suggest authors to consider adding in more (non image related) features into the forecast process

---

### Decision · Program_Chairs · 2019-05-06
**Acceptance Decision**

Accept